# Investigation of the Impact of Neutron Irradiation on SiC Power MOSFETs Lifetime by Reliability Tests

**DOI:** 10.3390/s21165627

**Published:** 2021-08-20

**Authors:** Fabio Principato, Giuseppe Allegra, Corrado Cappello, Olivier Crepel, Nicola Nicosia, Salvatore D′Arrigo, Vincenzo Cantarella, Alessandro Di Mauro, Leonardo Abbene, Marcello Mirabello, Francesco Pintacuda

**Affiliations:** 1Department of Physics and Chemistry—Emilio Segrè (DiFC), Palermo University, Viale Delle Scienze, Ed. 18, 90128 Palermo, Italy; leonardo.abbene@unipa.it (L.A.); marcello.mirabello@unipa.it (M.M.); 2STMicroelectronics, Stradale Primosole 50, 95121 Catania, Italy; giuseppe.allegra@st.com (G.A.); corrado.cappello@st.com (C.C.); nicola.nicosia@st.com (N.N.); salvatore.darrigo@st.com (S.D.); vincenzo.cantarella@st.com (V.C.); alessandro.di-mauro@st.com (A.D.M.); francesco.pintacuda@st.com (F.P.); 3Central Airbus R&T, LDC Airbus SAS, Oliver Crepel E177, 14 rue Gabriel Clerc, CEDEX, 31707 Blagnac, France; olivier.crepel@airbus.com

**Keywords:** power device reliability, failure in time, silicon carbide, neutron beam, single event burnout

## Abstract

High temperature reverse-bias (HTRB), High temperature gate-bias (HTGB) tests and electrical DC characterization were performed on planar-SiC power MOSFETs which survived to accelerated neutron irradiation tests carried out at ChipIr-ISIS (Didcot, UK) facility, with terrestrial neutrons. The neutron test campaigns on the SiC power MOSFETs (manufactered by ST) were conducted on the same wafer lot devices by STMicroelectronics and Airbus, with different neutron tester systems. HTGB and HTRB tests, which characterise gate-oxide integrity and junction robustness, show no difference between the non irradiated devices and those which survived to the neutron irradiation tests, with neutron fluence up to 2× 10${}^{11}$ (n/cm${}^{2}$). Electrical characterization performed pre and post-irradiation on different part number of power devices (Si, SiC MOSFETs and IGBTs) which survived to neutron irradiation tests does not show alteration of the data-sheet electrical parameters due to neutron interaction with the device.

## 1. Introduction

Silicon carbide (SiC) power MOSFETs are now used in several applications such as automotive and avionic, due to their better electrical performance respect to silicon counterpart. Concerning the device reliability of SiC power MOSFETs, the ruggedness of these devices against the impact of neutrons generated from cosmic ray interactions with the atmosphere is now a big concern for its impact on operating life of power systems [1,2]. The atmospheric neutron interaction with power MOSFETs operating at high-voltages may cause Single Event Burnout (SEB), which results in catastrophic device failure [3]. When the neutrons strike the lattice atoms realise energy and create electron-hole pairs. If this interaction occurs in a region where is present an high electric field, it could destroy the device. The turn-on of the parasitic bipolar junction transistor present in the MOSFET structure due to the neutron interaction is the conventional mechanism to explain the SEB phenomenon in silicon power MOSFETs [3,4], and it is sometimes invoked also to explain the SEB in SiC MOSFET [5], although similar SEB tolerance has been found in SiC diodes [6]. Some authors report also gate damage in power MOSFET’s after neutron exposure, which for some devices results in a complete gate rupture (SEGR) [7]. Failure mechanisms in high-voltage power semiconductor devices due to the cosmic radiation limit the maximum operating DC blocking voltage, thus accelerated terrestrial neutron irradiations were usually performed on commercial power MOSFETs to estimate the failure in time (FIT) (1 FIT = 1 failure/billion hours) at different de-rated values of their maximum rated blocking voltages [4,5,8,9,10,11]. The atmospheric neutron flux at sea level is 13 (n/cm${}^{2}$ h) for neutron energies higher than 10 MeV and it increases with the altitude with a peak at approximately 20-km. For example, at 12 km of altitude, which is the worst case scenario for avionic applications, the neutron flux is 6000 (n/cm${}^{2}$ h) [12]. The devices that work for 10${}^{9}$ hours are exposed at a neutron fluence of $1.3\times 10^{10}$ (n/cm${}^{2}$) at sea level and $6.0\times 10^{12}$ (n/cm${}^{2}$) at 12 km of altitude. Hence, the FIT constrains are more stringent for power devices used in avionic applications. Generally, it is necessary to achieve low FIT values especially when many power devices are used in modules which have to satisfy industry qualification standards. For example, in automotive applications it is necessary to achieve device failure rate below 10 FIT in order to have module failure rate below 100 FIT [13].

During accelerated neutron tests the devices can be exposed to high neutron fluence. This fluence depends on the device FIT value and on the number of the tested samples. The neutron fluence increases with FIT decreasing and for the same FIT value increases with the altitude and with decreasing the number of tested devices. Moreover, due to exponential dependance of the FIT on the bias drain voltage [14], the neutron exposure during the neutron tests increases with lowering the drain voltage under irradiation.

In addition to SEB and SEGR mechanisms, generally, neutrons can cause displacement damages in semiconductor devices, which is due to non-ionizing effects when the neutrons traversing a device. This effect is associated the non-ionizing energy loss rate (NIEL) concept, which quantifies that portion of the energy lost by an incident particle that goes into displacements [15]. Neutron displacement damage is well known in bipolar transistors and it is used to realize neutron dosimeters. In [16] tests performed on different kinds of commonly used bipolar transistors prove that none are sensitive to neutron exposure fluence below 1 × 10${}^{12}$ (n/cm${}^{2}$). The effects displacement damage on low-voltage vertical power MOSFETs were investigated in [17] by exposing the devices to proton irradiations. The results show defects introduced by displacement damage can significantly affect the carrier mobility of the devices when the proton dose reaches ≈$10^{15}$ (n/cm${}^{2}$) 1-MeV neutron equivalent fluence.

The issue of the degradation of power devices which do not fail during neutron irradiation was already investigated by some authors [7,11]. In [7] accelerated neutron tests were performed on commercial SiC power MOSFETs with different structures. They found that some power devices show degradation with respect to the pristine device. In [11] the authors show that silicon MOSFETs and IGBTs after irradiation with 1.9 MeV mono-energetic neutron beam, produce threshold voltage and the leakage gate current degradation.

The aim of this work is to investigate not only degradation phenomena but also the impact on the reliability of the devices of SiC power MOSFETs do not fail during a neutron irradiation run. In [8] we performed an electrical characterization to evaluate the gate-oxide integrity on Si and SiC power devices which survive the neutron tests. In this work we extend this investigation by means a more complete electrical characterization of all survived test vehicles irradiated at Chip-Ir facility during many neutron test campaigns, which include also those reported in [8]. These neutron test campaigns were performed to increase the statistics and to evaluate neutron performance at avionic altitude, by using different neutron tester systems as well.

We also performed the statistical analysis of the failed devices, in order to verify if the neutron fluence to failure has a cumulative distribution function which follows the exponential distribution, which usually is a-priori assumed in failure rate calculation.

Moreover, for the GEN3 (SCT130N120G3) SiC MOSFETs (STMicroelectronics) we further investigated the impact on the device lifetime of the neutron exposure by performing High Temperature Reverse Bias (HTRB) and High Temperature Gate Bias (HTGB) reliability tests. The investigated samples of GEN3 SiC MOSFETs were irradiated at the ChipIr facility with two different neutron test systems to estimate the FIT value at sea level and avionic altitude. The whole survived devices were electrical characterized before and after irradiation by measuring all static data-sheet parameters.

## 2. Experimental

All test vehicles (see Table 1 of [8] ) were irradiated with neutrons at the ChipIr beamline of the ISIS spallation source at the Rutherford Appleton Laboratory, UK facility. These devices were irradiated in different neutron test campaigns preformed by STMicroelectronics and Airbus and according to the JEDEC JEP151 procedure [18]. Some results of the accelarated neutron testing by ST are reported in [8]. These devices were tested to determine the sea-level failure rate at different drain bias and with zero gate-source voltage by using a neutron tester system developed by ST. The devices survived the irradiation test were exposed to neutron fluence in the range 8 × 10${}^{9}$–3 × 10${}^{10}$ (n/cm${}^{2}$). Samples of the GEN3 SiC power MOSFETs, from the same wafer lot, were also irradiated at the ChipIr facility by the Airbus company to test the neutron ruggedness of these devices at avionic altitude (12 km). The irradiated samples were biased and controlled with the Airbus test bench, capable to test up to 3 kV and monitor the gate and drain current of the MOSFET. The GEN3 SiC MOSFETs samples were irradiated at $V_{DS}=$ 800 and 850 V with fluence up to 2 × 10${}^{11}$ (n/cm${}^{2}$). These bias conditions are those used in applications of this power device. The FIT results of the irradiated samples are shown in Table 1.

### 2.1. Electrical Characterization

In Pre and Post-Stress Electrical Test, data-sheet electrical parameters were measured to monitor the electrical performance of the MOSFETs. In Table 2 are shown the symbols and test conditions of some electrical paramenters of the power MOSFET. Some parameters are measured at different conditions (to simplify, the table shows just one condition). The electrical parameters were measured with the Transistor Tester System for Power Discrete Devices TESEC Model 800I-TT.

The electrical measurements performed with the TESEC instrument does not allow to reach a good accuracy for the measurement of the current with values below ≈100 nA. We measured both the gate and drain currents with better accuracy than that of the TESEC instrument, by using the SMU Keithely 2635. We performed measurements of the sub-threshold and Flower-Nordheim curves (i.e., $I_{DS}-V_{GS}$ at $V_{DS}=2$ V and $I_{GS}-V_{GS}$ at $V_{DS}=0$ V, respectively) of the GEN3 SiC MOSFET before neutron irradiation and for the samples which survived the test. It is known that SiC MOSFET’s suffer of the threshold-voltage instability, due mainly to the defects at the SiC/SiO${}_{2}$ interface [19]. The threshold-voltage instability effect is the shifting of threshold $V_{TH}$ back and forth in response to an applied gate-bias stress. Hence, the $I_{DS}-V_{GS}$ curve at $V_{DS}$ constant depends on the sweeping mode of the $V_{GS}$ voltage. To minimize this effect we use a preconditioning technique, which brings the interface into a well-defined and reproducible charge state before the measurement, by applying two consecutive gate bias voltage pulses, with a few seconds of duration [20].

### 2.2. HTGB and HTRB Tests

HTRB and HTGB tests were performed according to the test conditions for qualification of discrete semiconductors resported in [21]. During the tests, the DUTs were placed in an environmental chamber which provided a constant temperature of 200 °C and in the meantime they were bias stressed by applying a DC voltage. The following stress bias conditions were used:HTRB $V_{GS}$ = 0 V, $V_{DS}$ = 1200 VHTGB1 $V_{GS}$ = +22 V, $V_{DS}$ = 0 VHTGB2 $V_{GS}$ = −10 V, $V_{DS}$ = 0 V

All the devices were stressed under the described condition for 1000 h (6 weeks). Three measurements steps were performed after 168, 500 and 1000 h. These tests were performed on 59 samples of GEN3 SiC MOSFETs split in pristine no irradiated (No Irr) 30 samples, 16 samples irradiated by STMicroelectronics (ST) and 13 samples by Airbus (AB).

## 3. Results

### 3.1. Electrical Characterization of the Survived Samples

The electrical characterization with the automatic TESEC equipment were performed on the samples before they were irradiated at ChipIr facility by ST and Airbus companies. At the same time, the same characterization was also performed on 10 golden samples of GEN3 SiC MOSFET, from the same wafer lot as the irradiated devices. The results of the electrical parameters on the survived show that the all parameters do not suffer of any degradation. The drift of the parameters of the survived samples respect to the values measured before the irradiation tests are of the same magnitude of those of the golden samples. For example, the the mean of the percentage drift of the threshold $V_{TH}$ at $I_{DS}=$ 1 mA of the survived GEN3 MOSFETs is 0.75%, whereas that of the golden samples is 0.67%, for the leakage $HVI_{DSS}$ at $V_{DS}=$ 1.2 kV the drifts of both the survived devices and golden samples are of order of magnitude of a few nA. These drift values are comparable with the accuracy limit of the measurement equipment.

In the Figure 1a are shown the $I_{DS}-V_{GS}$ at $V_{DS}=2$ V curves of three samples of the GEN3 SiC MOSFETs which survived the neutron irradiation. For each sample, the curves measured before and after irradiation are shown. In the same figure are shown two measurements of the same curves of a golden sample of the GEN3 SiC MOSFET, which was used to verify the reproducibility of the measurements. We note that the second curve is slightly shifted with respect to the first one. This shift can be due threshold−voltage instability of the SiC MOSFETs [20]. The preconditioning technique used to reduce this effect seems to work in the case of consecutive measurements made in a short period of time. We note that the sub−threshold curves of the three survived samples show after irradiation the same slightly shift of the non-irradiated golden sample. Hence, we conclude that the neutron irradiation does not alter the sub−threshold curves of GEN3 SiC MOSFETs. In the Figure 1b are shown the $I_{GS}-V_{GS}$ at $V_{DS}=0$ V curves of three samples of the GEN3 SiC MOSFETs, measured for positive values of the gate-source voltage. The value of the onset voltage of the Fowler-Nordheim regime is ≈27 V. We note that these curves are not modified after neutron exposure. This demonstrates that the neutron interaction with the crystal lattice do not cause gate-oxide damage. This behaviour is different from that of Heavy-Ions, which can induced gate-oxide degradation as a secondary effect of the interaction of the ions within the Body-Drain Junction creating hot-carriers (holes) which are injected into the gate-oxide causing its degradation [22].

The same electrical characterization was also performed in IGBTs, Si and SiC MOSFET devices, which survived neutron irradiation. For each part-number golden sample devices were used. The results show that for all Si and SiC devices the drifts of the measured parameters are of the same order of magnitude of the respective golden samples.

### 3.2. Weibull Analysis of the Irradiation Failures

To investigate statistical properties of the failure events in power electronic devices biased at high voltage due the neutron interaction, we plot the cumulative failure distribution $F\left(\phi \right)$, where ϕ is the neutron fluence to failure. We assume that $F\left(\phi \right)$ obeys to the two parameters Weibull model
$$F\left(\phi \right)=1-e^{-{\left(\phi /\eta \right)}^{\beta }}$$
where β is the shape parameter and η is the scale parameter. β close to 1 is in agreement with random failures implying that η is the mean fluence to failure. In this case, the FIT parameter, at sea level, is related to scale factor by FIT = $13\times 10^{9}/\eta$. The aim of this study is to determine the shape factor β, in order to verify if the failures follow an exponential distribution and therefore the FIT value does not depend on the time exposure. The estimation of the Weibull parameters is a critical issue, and several methods exist to determine these parameters [23]. Several probability estimators of the *i*-th failure $F_{i}$ has been used and we choose the Hazen estimator $F_{i}=\left(i-0.5\right)/n$, where *n* is the number of failures. This estimator gives the least bias [24]. For the β parameter we use the maximum likelihood estimator, which demonstrates, between the common approaches in β estimation, the narrowest distribution for all sample sizes [25].

The purpose of this study is to check if the failure distribution function is comparable for different device types and for the same device under different bias conditions. In particular, we investigate if the shape parameter of the Weibull distributions is close to 1. In Figure 2 are shown the Weibull plots of two different SiC power MOSFET’s (Figure 2a,b), Si N-MOSFET Figure 2c) and IGBT Figure 2d as a function of the neutron fluence ϕ.

In Table 3 are shown the values of the shape parameter β for the the Weibull plots of the Figure 2. In this table are also indicated the extreme values of the 95% confidence interval (CI) of the estimated $\hat{\beta }$ value. We note that the estimated values of the shape parameters are close to 1. This result is showed by all tested power devices at different drain bias voltages during irradiation.

### 3.3. HTGB and HTRB Tests on Survived GEN3 SiC MOSFETs

In Table 4 are shown the results of the HTGB and HTRB tests performed on the survived and pristine samples of the GEN3 SCT130N120G3 SiC MOSFET devices.

We note that no failures occurs after 1000 h of HTRB and both HTGB tests for pristine and survived devices irradiated by ST and Airbus. In Table 5, Table 6 and Table 7 are shown the measured values of the main parameters of the power devices before and after each reliability step. The tables show the median value of the electrical parameters and the maximum value of the drift after the tests respect to those measured at 0 hour. Moreover, in the tables are indicated the maximum drift acceptable, according to the specifications of [21]. Test failures are defined as devices exhibiting values out the data-sheet specification or devices not remaining within ±20% of the initial reading of each test with the exception of leakage limits which have to not exceed 5 times the initial value. All the tested devices have electrical parameters drift well below these constrains. Concerning the gate leakage currents we note that the devices have values during all the test steps below 100 nA and the AEC spec. asserts that in this case the tester accuracy prevent a post stress analysis to initial reading. Based on these reliability results, we can assert that there are not difference in the statistical estimators of the static electrical parameters between the survived to neutron irradiation and pristine samples.

To highlight any differences between pristine and irradiated devices of the GEN3 SiC MOSFET we show in the Figure 3 the time evolution of $V_{TH}$, $BV_{DSS}$, $V_{DSon}$ and $V_{FECS}$ parameters during the HTRB stress. In the Figure 4 are shown the same parameters during the HTGB2 stress. The dots indicate the median values and the error bars the minimum and maximum of the data set. We note that there are no marked differences between the parameters during the HTRB stress between the pristine and survived samples. The initial (t = 0 h) median value differences between the three sets of the GEN3 samples is due to the usual spreading process, even if all devices come from the same production lot. Moreover, the spread of outliers does not undergo a significant variation during the stress for the whole parameters in all investigated devices. The same behaviour have these parameters during the HTGB1 stress. Hence, we conclude that the neutron irradiation does not cause neither parameter degradation of the SiC MOSFETs nor alter its lifetime.

## 4. Discussion

In [11] was investigated the degradation effects of silicon Super-Junction MOSFETs and IGBTs after irradiation, at $V_{GS}$ = 0 V and different % values of the rated breakdown voltage, with 1.9 MeV mono-energetic neutron beam and up to fluence 7.38 $\times \;10^{12}$ (n/cm${}^{2}$). The results of [11] show that all devices, which did not result in failure, present charge trapping at the silicon/oxide interface, which causes the $V_{TH}$ shift in the 0.2–0.4 V range, and displacement damage visible in both the increasing of the leakage currents and the shift of the $BV_{DSS}$ value only for the devices irradiated at 100% $BV_{DSS}$. The authors conclude that fast neutrons at energies of 1.9 MeV can induce both TID and displacement damage in power semiconductors.

These results differ from those found in our investigation, in which neither TID nor displacement damage is present in irradiated samples. The main difference which could explain the lack of degradation in our device is the maximum neutron fluence values, which in our case is roughly one order of magnitude lower.

In [7] accelerated terrestrial neutron irradiations were performed at the ChipIr facility on different commercial SiC power MOSFETs with planar, trench, and double-trench architectures. Concerning the drain leakage current and the breakdown voltage they observed degradation induced by the neutron exposure in both planar and trench-power MOSFETs devices. All devices were tested with fluence up to $2.8\times 10^{10}$ n/cm${}^{2}$. They also performed post irradiation gate stress (PIGS) test to study the effect of latent damage on the gate oxide integrity. The results shown the gate damage increased after the PIGS only for devices with the gate partially ruptured after the neutron exposure. The authors conclude that the devices with planar-gate architecture exhibit a partial gate rupture mechanism, similar to degradation induced by heavy ions exposure. Hence, the results reported in [7] differ from those found in our investigated devices. In our case, concerning the SiC GEN3 MOSFET, both the Fowler-Nordheim curves and the HTGB tests, performed in survived samples, do not reveal degradation in the gate oxide, even up to neutron fluences one order of magnitude higher respect to that used in [7].

We note that some power MOSFETs tested in [7] were irradiated with different drain bias voltage $V_{DS}$ up to 92% of the maximum rated voltage $BV_{DSS}$. They observe that trench technology is more sensitive to degradation than the planar one. Moreover, we note that among the tested power devices of different manufactured, the ST power MOSFETs do not present any degradation. Values of drain bias voltage greater than 70% of the rated drain voltage are not used in power applications. Although correlation between the $V_{DS}$ bias voltage and the degradation of the devices were not investigated in [7], these high values of the drain bias voltage could have triggered degradation phenomena. Indeed, in [26] several power MOSFETs operating at maximum-rated drain voltage under broad-spectrum neutron beam, leakage current increased due to neutron-induced damage.

Our results are in agreement with those obtained in [14,27], where SiC power MOSFETs of different vendors irradiated with terrestrial-like spectrum neutrons which do not fail during irradiation do not show electrical degradation up to neutron fluence of 4 $\times \;10^{10}$ (n/cm${}^{2}$). Moreover, in [14] PIGS measurements were taken for SiC power MOSFETs that did not fail during irradiations and those survived irradiation also survive the PIGS test.

The HTGB stress tests performed in the GEN3 SiC power MOSFETs shows that the survived devices to neutron irradiation do not suffer of any gate oxide degradation and by any significant change of the number of charge traps at and near SiC/SiO${}_{2}$ interface. The same devices survived the HTRB tests without significant variation of their electrical parameters, demonstrating that the neutron interaction with sensitive volume of the devices, if does not trigger SEB, does not alter the junction integrity, the crystal defects and the ionic-contamination level.

The absence of any degradation in the investigated power devices which survived to neutron irradiation demonstrated in this paper, at least up to neutron fluence ≈10${}^{11}$ (n/cm${}^{2}$), confirms that the physical mechanisms for nucleon-radiation induced failure in power electronic devices stipulates a constant hazard rate, which allows to model the fluence-to-failure random variable with the exponential distribution function and to evaluate the device failure rate corresponding to the device exposure to cosmic radiation [18]. Our results on the estimated failure distribution function $F\left(\phi \right)$ of the tested SiC and Si power MOSFET’s and IGBT’s, irradiated at different bias conditions, show that $F\left(\phi \right)$ can be described by a two-parameter Weibull distribution. The 95% confidence intervals of the estimated shape parameter $\hat{\beta }$, which provides information about the precision of the point estimate, include in all cases the value 1. Hence, although the number of failed samples used to estimate the Weibull parameters is typically not great to get small coefficients of variation, due mainly to economic reasons, our Weibull analysis gave us some confidence that $F\left(\phi \right)$ could follow the exponential distribution function.

## 5. Conclusions

Si and SiC power MOSFETs and IGBTs by STMicroelectronics were irradiated with terrestrial-like spectrum neutrons at ChipIr-ISIS (Didcot, UK) facility, with two different neutron tester systems by ST and Airbus. Power devices which survived to the neutron irradiation tests, were electrical characterized pre and post-irradiation. The results show that all power devices do not suffer of any degradation in the electrical parameters when irradiated with neutron fluence in the range $1\times 10^{9}$–$2\times 10^{11}$ and at different bias conditions with $V_{GS}=0$ V. This result was obtained by investigating a few hundred of survived device samples of different power technologies and materials.

The GEN3 SiC MOSFETs, survived to neutron irradiation, were also evaluated by the reliability tests High Temperature Gate Bias (HTGB) and High Temperature Reverse Bias (HTRB). All irradiated GEN3 SiC power MOSFETs which survived to neutrons pass even these reliability tests, at least up to neutron fluence of $2\times 10^{11}$. Although the sample size of the devices stressed in the reliability tests is limited, this result demonstrates that the neutron interaction with the power devices can induce a destructive event (SEB) otherwise has no impact on the device lifetime. This result is also confirmed by the Weibull analysis performed on the failed devices, which highlights that the interaction of neutrons with power devices under high bias voltage is a memoryless process. 

## Figures and Tables

**Figure 1 sensors-21-05627-f001:**
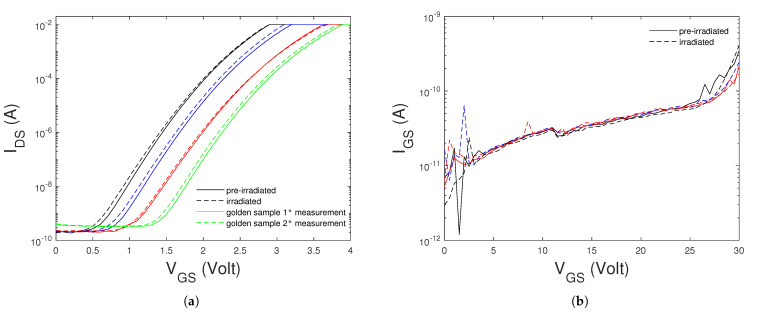
Sub−threshold (**a**) and Fowler-Nordheim (**b**) curves of three samples of the GEN3 SiC MOSFET before and after neutron irradiation. Two sub−threshold curves of a golden smaple are also shown.

**Figure 2 sensors-21-05627-f002:**
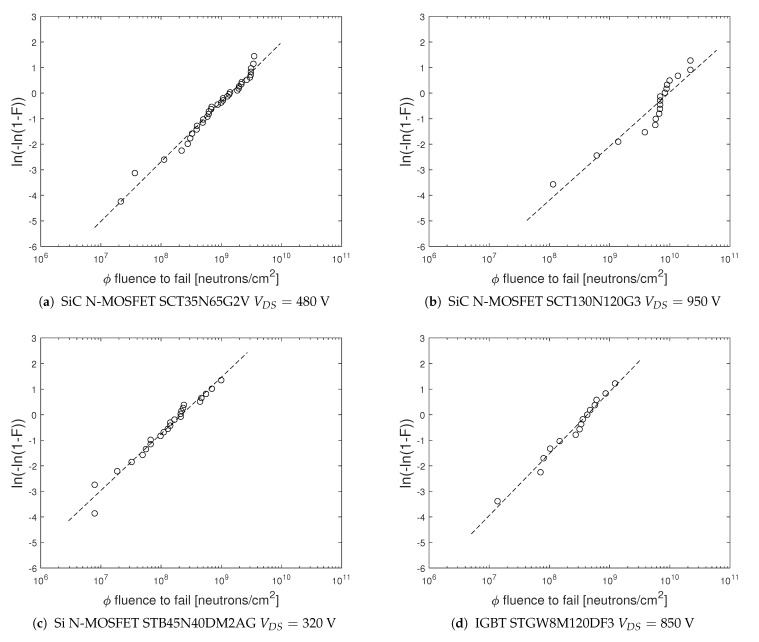
Weibull−plots of failed SiC and Si power MOSFETs as a function of neutron fluence.

**Figure 3 sensors-21-05627-f003:**
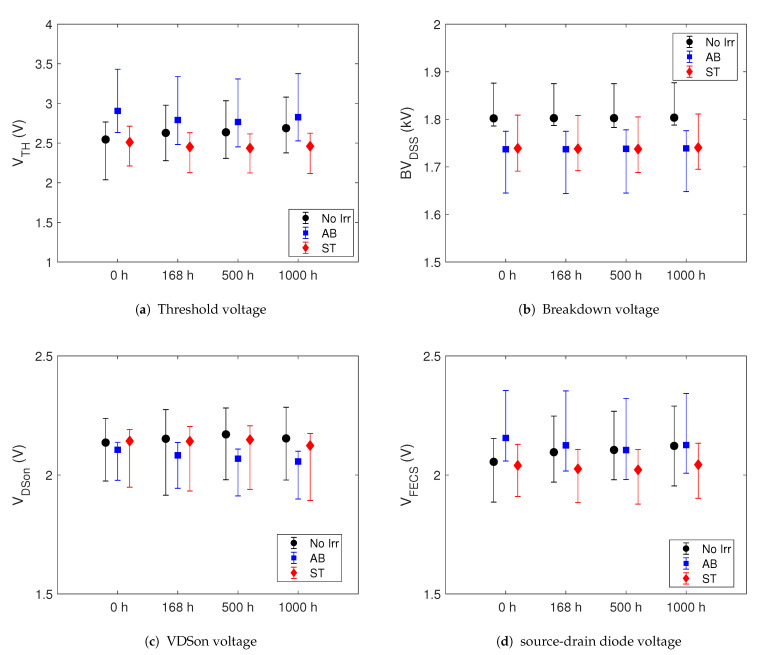
Shift of the median of some electrical parameters during HTRB stress of GEN3 SiC MOSFETs. The error bars indicate the minimum and maximum value.

**Figure 4 sensors-21-05627-f004:**
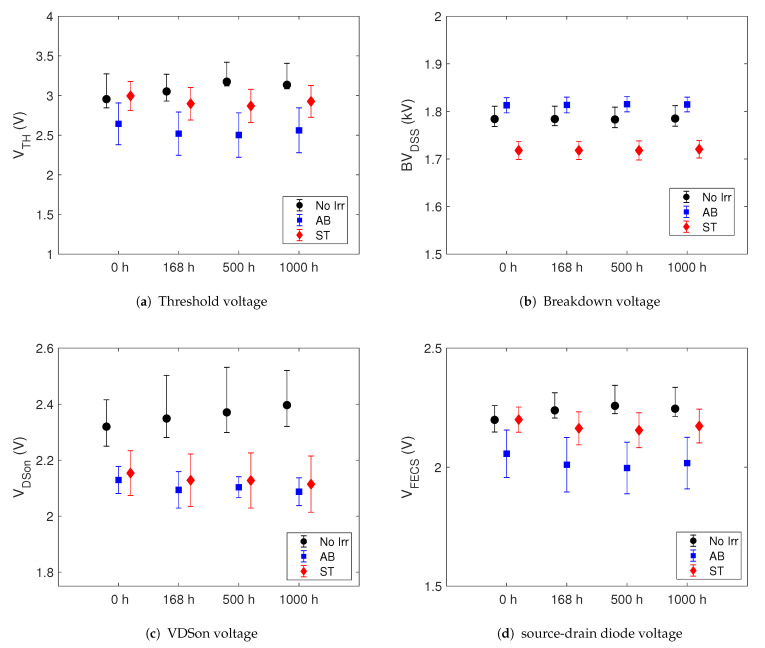
Shift of the median of some electrical parameters during HTGB2 stress of GEN3 SiC MOSFETs. The error bars indicate the minimum and maximum value.

**Table 1 sensors-21-05627-t001:** The FIT values of the GEN3 SiC MOSFET tested by Airbus.

$V_{\mathrm{DS}}$(V) under Irradiation (V)	FIT Sea Level [95% CI Bounds]	FIT 12 km [95% CI Bounds]
800	0.01 [0.006–0.032]	4.6 [2.81–14.72]
850	0.09 [0.053–0.143]	42 [24.73–65.78]

**Table 2 sensors-21-05627-t002:** Symbols and test conditions of some electrical parameters of the power MOSFETs.

Symbol	Unit	Parameter	Condition at $T_{j}$ = 25 °C
$I_{SGS}$	nA	gate leakage current	$V_{GS}=-$10 V, $V_{DS}=$ 0 V
$I_{GSS}$	nA	gate leakage current	$V_{GS}=+$22 V, $V_{DS}=$ 0 V
$V_{TH}$	V	gate-source threshold voltage	$I_{DS}=$ 1 mA, $V_{G}=V_{D}$
$I_{DSS}$	μA	drain leakage current	$V_{DS}=$ 1.2 kV, $V_{GS}$ = 0 V
$BV_{DSS}$	V	drain-source breakdown voltage	$I_{DS}=$ 1 mA, $V_{GS}$ = 0 V
$V_{DSon}$	V	drain-source on-state voltage	$I_{DS}=$ 120 A, $V_{GS}$ = 18 V
$V_{FECS}$	V	source-drain diode voltage	$I_{DS}=$ 10 A, $V_{GS}$ = 0 V

**Table 3 sensors-21-05627-t003:** Shape parameter $\hat{\beta }$ for some power devices with the lower and upper limits of the 95% confidence interval.

Power Device	${\mathrm{BV}}_{\mathrm{DSS}}$ (V)	$V_{\mathrm{DS}}$ (V) (under Irradiation)	$\hat{\beta }$	Lower β 95% CI	Upper β 95% CI
SiC-SCT35N65G2V	650	480	1.1	0.86	1.50
SiC-SCT130N120G3	950	1200	1.2	0.87	1.50
Si-STB45N40DM2AG	325	400	0.95	0.70	1.30
IGBT-STGW8M120DF3	850	1200	1.20	0.79	1.80

**Table 4 sensors-21-05627-t004:** HTRB and HTGB tests results of the GEN3 SiC MOSFETs.

					Failures/Samples		
**Test**	**Standard Reference**	**Conditions**	**n° Samples**	**Step**	**No Irr**	**ST**	**AB**
HTRB	MIL-STD-750-1M1038	$T_{j}$ = 200 °C, BIAS = 1200 V	23	168 h	0/10	0/6	0/7
				500 h	0/10	0/6	0/7
				1000 h	0/10	0/6	0/7
HTGB1	JESD22A-108	$T_{j}$ = 200 °C, BIAS = +22 V	17	168 h	0/10	0/4	0/3
				500 h	0/10	0/4	0/3
				1000 h	0/10	0/4	0/3
HTGB2	JESD22A-108	$T_{j}$ = 200 °C, BIAS = −10 V	19	168 h	0/10	0/6	0/3
				500 h	0/10	0/6	0/3
				1000 h	0/10	0/6	0/3

**Table 5 sensors-21-05627-t005:** Drifts of some GEN3 SiC MOSFET electrical parameters for the HTRB test.

HTRB							
Parameter	Median				Max Drift	Spec. Drift	Datasheet Limit
	0 h	168 h	500 h	1000 h			
$I_{GSS}$ @ $V_{GS}=$ −10 V	<10 nA	<10 nA	<10 nA	<10 nA	-	5 times initial value	<100 nA
$I_{GSS}$ @ $V_{GS}=$ +22 V	<10 nA	<10 nA	<10 nA	<10 nA	-	5 times initial value	<100 nA
$V_{TH}$ @ $I_{DS}=$ 1 mA	2.58 V	2.48 V	2.47 V	2.46 V	<5%	<20%	>1 V, <6 V
$I_{DSS}$@ $V_{DS}=$ 1.2 kV	0.44 μA	0.40 μA	0.38 μA	0.32 μA	-	5 times initial value	<10 μA
$BV_{DSS}$ @$I_{DS}=$ 1 mA	1.73 kV	1.73 kV	1.73 kV	1.74 kV	<2%	<20%	>1.2 kV, <1.9 kV
$V_{DSon}$ @ $I_{DS}$ = 120 A $V_{GS}=$ 18 V	2.16 V	2.14 V	2.15 V	2.12 V	<2%	<20%	>1.5 V, <4.14 V
$V_{FECS}$ @ $I_{DS}=$ 6 A	2.06 V	2.04 V	2.04 V	2.04 V	<1%	<20%	<5 V

**Table 6 sensors-21-05627-t006:** Drifts of some GEN3 SiC MOSFET electrical parameters for the HTGB1 test.

HTGB1							
**Parameter**	**Median**				**Max Drift**	**Spec. Drift**	**Datasheet Limit**
	**0 h**	**168 h**	**500 h**	**1000 h**			
$I_{GSS}$ @ $V_{GS}=$ −10 V	<10 nA	<10 nA	<10 nA	<10 nA	-	5 times initial value	<100 nA
$I_{GSS}$ @ $V_{GS}=$ +22 V	<10 nA	<10 nA	<10 nA	<10 nA	-	5 times initial value	<100 nA
$V_{TH}$ @ $I_{DS}=$ 1 mA	2.76 V	2.92 V	3.02 V	3.02 V	<5%	<20%	>1 V, <6 V
$I_{DSS}$@ $V_{DS}=$ 1.2 kV	0.30 μA	0.29 μA	0.29 μA	0.06 μA	-	5 times initial value	<10 μA
$BV_{DSS}$ @$I_{DS}=$ 1 mA	1.78 kV	1.78 kV	1.78 kV	1.74 kV	<2%	<20%	>1.2 kV, <1.9 kV
$V_{DSon}$ @ $I_{DS}$ = 120 A $V_{GS}=$ 18 V	2.32 V	2.29 V	2.35 V	2.47 V	<2%	<20%	>1.5 V, <4.14 V
$V_{FECS}$ @ $I_{DS}=$ 6 A	2.16 V	2.20 V	2.23 V	2.08 V	<1%	<20%	<5V

**Table 7 sensors-21-05627-t007:** Drifts of some GEN3 SiC MOSFET electrical parameters for the HTGB2 test.

HTGB2							
**Parameter**	**Median**				**Max Drift**	**Spec. Drift**	**Datasheet Limit**
	**0 h**	**168 h**	**500 h**	**1000 h**			
$I_{GSS}$ @ $V_{GS}=$ −10 V	<10 nA	<10 nA	<10 nA	<10 nA	-	5 times initial value	<100 nA
$I_{GSS}$ @ $V_{GS}=$ +22 V	<10 nA	<10 nA	<10 nA	<10 nA	-	5 times initial value	<100 nA
$V_{TH}$ @ $I_{DS}=$ 1 mA	2.94 V	3.20 V	3.22 V	3.21 V	<5%	<20%	>1 V, <6 V
$I_{DSS}$@ $V_{DS}=$ 1.2 kV	0.40 μA	0.41 μA	0.41 μA	0.42 μA	-	5 times initial value	<10 μA
$BV_{DSS}$ @$I_{DS}=$ 1 mA	1.76 kV	1.76 kV	1.76 kV	1.76 kV	<2%	<20%	>1.2 kV, <1.9 kV
$V_{DSon}$ @ $I_{DS}$ = 120 A $V_{GS}=$ 18 V	2.27 V	2.36 V	2.40 V	2.40 V	<2%	<20%	>1.5 V, <4.14 V
$V_{FECS}$ @ $I_{DS}=$ 6 A	2.20 V	2.30 V	2.31 V	2.30 V	<1%	<20%	<5V

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
