# Peer review of "Investigation of the Impact of Neutron Irradiation on SiC Power MOSFETs Lifetime by Reliability Tests"

_sensors, 2021, doi:10.3390/s21165627_

Round 1

Reviewer 1 Report

Dear authors,

thank you for providing this interesting study. From my point of view, it provides interesting and relevant findings on the behavior of SiC devices under neutron iradiation, which is important for users of this technology.  Your study is well introduced and well reasoned. The experiments are clearly described and the results are shown in an adequate manner. A discussion of the experimental results is provided and conclusions are drawn.

There are several possibilities to improve the readability, where I provide some suggestions to you. Furthermore, I see potential in adapting your conclusions better to the experimental findings, since some statements seem to be rather strong to me and some experimental findings need to be interpreted in a more differentiated way.

Specific comments:

Title:    change “releability” to reliability

line 3:        change “carry out” to carried out

32-34: “The atmospheric neutron flux at sea level is 13 (n/cm2 h) for neutron energies higher than 10 MeV 32 and it increases with the altitude with a peak at approximately 20-km. For example, at 2.74 Km of 33 altitude the neutron flux is 9 time higher than on sea level.”
Please give a reference for this. It would be furthermore helpful to have a neutron flux value for typical avionic heights (e.g. 10-20 km), the value given for an altitude of 2.74 km is not relevant for this application.

35:       What are typical requirements for FIT rates? (incl. reference, if possible)

67-72: It might be helpful to provide a brief table which shows the most relevant specifications of all tested device types.

87:       The applied fluence of 2E11 n/cm2 does not correspond to the fluence at an altitude of 12 km (cf. line 38/39). To which altitude does this fluence correspond?

Table 1:           please indicate the number of samples tested to obtain the respective FIT numbers. Is it also possible to provide the uncertainties of the received FIT values?

Table 2:           This table does not display electrical parameters of the MOSFETs, but instead the test conditions under which the parameters have been determined. Please correct either the table or the table caption and the corresponding text in line 90.

107:    “AEC - Q101 - Rev – D1 specification” – please add a reference.

136:    “We note that these curves are not modified after neutron exposure.” Indeed, the irradiated curves are slightly shifted towards higher IDS for given VGS. This should be noted to be correct. This is of course only a small effect; but if there is a possible explanation, it should be given.

Fig. 1a+b:        It is unclear which line corresponds to which sample. Please use different line colours.

Fig. 2:              All sub-figures should have identical x and y axis ranges for a better comparability.

Section 3.2:     Please highlight that the purpose of this study is to check if the failure distribution function is comparable for different device types and testing conditions (if I understood correctly). Otherwise, the reader might be confused why four different sample types have been tested under four different conditions.

180-185:          You claim that there are no significant differences between irradiated and non-irradiated samples. However, the data shown in Fig. 3b (HTRB; breakdown voltage) and Fig. 4c (HTGB2; VDS,on) exhibits clear differences between these two sample types. Please comment on this and revise your conclusions accordingly. This also includes the discussion in section 4 (if applicable).

248-249:          “The results show that all power devices do not suffer of any degradation”. This seems to be a rather strong statement to me, when considering Figs. 1a, 3b and 4c where differences between irradiated and non-irradiated samples can be seen. Please reconsider this statement; or justify it.

Reviewer 2 Report

The authors conducted reliability tests on SiC devices which survived terrestrial neutron irradiation tests with fluences up to 10^11 n/cm^2 and showed that these devices did not present significant degradation in their electrical characteristics. The authors compared their results with those from previous researchers who observed degradation and indicated as important factors for this superior performance the use of a lower drain bias voltage and neutron fluence.  They also confirmed that a two-parameter Weibull distribution with shape parameter close to 1 is adequate to describe the observed failures.

The paper is technically sound and well organized, but it would benefit from a revision of its English. It has numerous minor errors, which do preclude its understanding, but are unacceptable in a scientific journal, for example,  "releability" in the title and "Flower-Nordheim" for the tunneling regime.

Author Response

Dear Reviewer,

thank you for your suggestions and comments. We hope that the English of revised manuscript has been improved. 

Round 2

Reviewer 1 Report

Dear authors,

thank you for revising your manuscript according to my suggestions. 

I recommend publishing the revised version of your manuscript which you have provided.

Kind regards.